# New Lead Schiff Bases Predominantly Mediate Vasorelaxant Activity Through α_1_ Receptor Blocking Activity

**DOI:** 10.3390/biom15050611

**Published:** 2025-04-23

**Authors:** Zakia Subhan, Niaz Ali, Abid Ullah, Wajid Ali, Muhammad Nabi, Syed Wadood Ali Shah

**Affiliations:** 1Department of Pharmacology, Institute of Pharmaceutical Sciences, Khyber Medical University, Hayatabad, Peshawar 25100, Khyber Pakhtunkhwa, Pakistan; drzakia.kims@kmu.edu.pk (Z.S.); dr.wajidali161@gmail.com (W.A.); muhammadnabi.ips@kmu.edu.pk (M.N.); 2Department of Pharmacology, College of Medicine, Shaqra University, Shaqra 11961, Saudi Arabia; 3Department of Pharmacy, Shaheed Benazir Bhutto University Sheringal, Dir Upper 18050, Khyber Pakhtunkhwa, Pakistan; abid@sbbu.edu.pk; 4Department of Pharmacy, University of Malakand Dir (Lower) at Chakdara, Chakdara 18800, Khyber Pakhtunkhwa, Pakistan; wadoodalishah@uom.edu.pk

**Keywords:** Schiff base, vasodilator, alpha one antagonist, Swiss target prediction, docking, verapamil

## Abstract

Schiff bases synthesized in our laboratory have demonstrated pain-relieving effects through both peripheral and central nervous system pathways. Considering that centrally acting analgesics often affect the muscle tone of the gastrointestinal tract (GIT) and related deep internal organs, this study was conducted to examine potential relaxant effects on blood vessels and GIT smooth muscles. The possible relaxant effects of Schiff bases (SB1 and SB2) on isolated rabbit aortic strips were evaluated. The experiments involved assessing their impact on contractions induced by 80 mM potassium chloride (KCL) and 1 µM norepinephrine (NE). Norepinephrine concentration response curves (N. ECRCs) were constructed in the absence and presence of three different concentrations of SB1 and SB2, using N. ECRCs as a negative control. Terazosin served as a standard α1 receptor blocker. Docking studies were employed to validate the mechanism of action for SB1 and SB2. The study outcomes suggest that SB1 is more potent than SB2, demonstrating lower EC_50_ values for NE-induced contractions in intact (5.50 × 10^−5^ ± 2.23 M) and denuded (5.81 × 10^−5^ ± 3.80 M) aortae. For NE-induced contractions, SB1 showed percent relaxation values of 48% and 41% in intact and denuded aortae, respectively. In comparison, SB2 exhibited values of 82.5% and 74%, showing that SB1 is more efficacious than SB2. The rightward shift of N. ECRCs for both SB1 and SB2 confirms their inhibition of α1 receptors. Additive effects of SB1 and SB2 were seen in the presence of verapamil (*p* < 0.0001). Docking analysis revealed that the compounds can properly bind to the target receptor Gq 1D (P25100). Findings show that both Schiff base SB1 and SB2 produce significant (*p* < 0.05) vasorelaxation via the α1 receptor blocking mechanism.

## 1. Introduction

Drug discovery is the process of finding potential new therapeutic compounds using computational, experimental, translational, and clinical models [1]. Thus, new drug development involves pre-clinical research, clinical trials, and regulatory clearance for launching drugs for treating various illnesses. Drug development is crucial for advancing healthcare and improving patient outcomes [2]. The process involves rigorous testing, evaluation, and collaboration between researchers, pharmaceutical companies, healthcare providers, and regulatory agencies [3]. The discovery process starts with identifying potential targets and lead compounds, which are then assessed for safety, efficacy, and pharmacokinetics. Only a few selective compounds pass the screening phases and may lead to development of life-saving medications [2,4].

These days, drug design is the creative process of discovering new drugs by applying data from biological targets using computational studies as well. Some lead molecules’ data predict the possible target sites for action. These predictions are then to be proved in either “in vitro” or “in vivo” models [5] for evidence-based results. In general, leads are then optimized to increase the affinity, selectivity, potency/efficacy, metabolic stability, oral availability, and reduction of adverse effects. The preclinical data are then translated into clinical studies and/or clinical trials that complete the process of new drug development [2,6]. We have targeted our SB1 and SB2 molecules for possible vasorelaxation and antispasmodic activity; therefore, we will discuss the relevant possible mechanisms.

The mechanism involved in the contractions of smooth muscles of blood vessels is through the entry of Ca^2+^ inside the cell. Extracellular calcium enters the cell through voltage-gated Ca^2+^ channels, and when an agonist interacts with an extracellular receptor, intracellular calcium is released from its internal storage [7]. Receptor and voltage-operated Ca^2+^ channels in the plasma membrane are closed during relaxation and thus reduce the quantity of Ca^2+^ that enters the cell [8]. Another pathway is through guanylyl cyclase catalyzing the generation of cyclic guanosine 3′,5′-monophosphate, triggered by nitric oxide, which is present in vascular smooth muscle. This cyclic nucleotide elicits vascular smooth muscle relaxation as it dephosphorylates the muscles [9].

Azomethine moiety (-N = CH-)-containing compounds are called Schiff bases. These have a variety of biological potentials. They exhibit a wide range of activities relevant to the medical and pharmaceutical industries [10], such as anti-oxidant, anti-microbial [11], anti-diabetic [12], anti-cancer, anti-depressant [13], anti-convulsant [14], anti-tubercular [15], anti-inflammatory, anti-pyretic [16], antihypertensive [17], and spasmolytic activity [18]. The structures of our lead molecules, SB1 and SB2, are shown in Figure 1A,B. Moreover, specific studies have shown that certain Schiff bases have a significant vasorelaxing effect, leading to a decrease in blood pressure and potential treatment for hypertension [19].

These findings suggest that Schiff bases could play a crucial role in improving cardiovascular health and may pave the way for the development of new drugs targeting vasorelaxation. This is going to open a new window for the role of Schiff bases in treating cardiovascular diseases as lead molecules. All information related to their synthesis and spectral details with purity are presented in the Appendix A as our target was the pharmacological screening of the lead molecules SB1 and SB2. Thus, the current work is aimed at discovering the underlying mechanisms of Schiff bases SB1 and SB2 on the smooth muscles of aortae and jejunal preparations using an in vitro animal model. These data are also supported by Swiss target prediction and molecular docking.

## 2. Materials and Methods

### 2.1. Study Setting

This study was conducted at the Pharmacology Department (Hakim Abdul Jalil Nadvi Herbal Research Centre) of the Khyber Medical University’s Institute of Pharmaceutical Sciences in Peshawar, Khyber Pakhtunkhwa, Pakistan.

### 2.2. Drugs and Chemicals

In the experiments, analytical grade chemicals were used. Acetylcholine (ACh) and norepinephrine were acquired from BDH, Poole, UK. Schiff bases (SB1 and SB2) were prepared by our research group [20]. Terazosin was purchased from Abbott Laboratories (Pakistan) Ltd. (Karachi, Pakistan). Verapamil was purchased from Searle Company Pak Limited (Karachi, Pakistan). On the same day of the experiments, suspensions and solutions were prepared.

### 2.3. Animals

The experiments utilized domestic rabbit breeds of both genders, weighing between 1.5 and 2.0 kg. These animals were housed for two weeks in a regulated environment at the Institute of Pharmaceutical Sciences, Khyber Medical University, Peshawar, Pakistan. Prior to experimental days, the rabbits were deprived of food overnight but had unrestricted access to water. Sick and pregnant animals were excluded from the study protocols. The Advanced Study and Research Board (ASRB) approved the study protocols, and ethical clearance was granted by the Ethical Research Board (ERB) of Khyber Medical University, Peshawar, Pakistan (Approval NO. KMU/IBMS/IRBE/5th Meeting/2023/9879-10).

### 2.4. Data Recording

Responses from aortic strips, both intact and denuded, were captured using Power-Lab (Model No. 0225 Pan Lab S1) transducers linked through bridge amplifiers. A four-channel PowerLab (Model No. 4/25 T, AD Instruments, Sydney, Australia) was used to record isolated vascular responses. For intestinal responses, a force transducer (Model No: MLT 0210/A Pan Lab S.I) was employed, connected via an FE 221 amplifier coupled to a four-channel PowerLab (Model No. 4/25 T) purchased from AD Instruments, Sydney, Australia.

### 2.5. Solutions

Normal Krebs’ solution consisted of the following ingredients (in mM): NaCl, 118.2; KCL, 4.7; KH_2_PO_4_, 1.3; MgSO_4_, 1.2; NaHCO_3_, 25.0; glucose, 11.7; and CaCl_2_, 2.5. The composition of Tyrode’s solution was as follows (in mM): NaCl, 136.9; KCl, 2.7; NaHCO_3_, 11.9; MgCl_2_, 1.1; glucose, 5.6; NaH_2_PO_4_, 0.4; and CaCl_2_, 1.8 (pH 7.4) [21].

#### Schiff Bases (SB1 and SB2) Had Poor Solubility in Distilled Water

To enhance their dispersion, the bases were suspended in 0.1% carboxy methyl cellulose (CMC) and 50 µL of dimethyl sulfoxide (DMSO). However, a negative control of 0.1% of CMC and 50 µL of dimethyl sulfoxide (DMSO) in deionized water was run to standardize the experiments. All suspensions and solutions were freshly prepared on the same days of experiments.

### 2.6. Effects of Schiff Bases (SB1 and SB2) on Spontaneous Rabbits’ Jejunal Preparations

Abdomens of overnight-fasted rabbits were opened as per our previously reported procedures. Their jejunums were removed and placed in petri dishes containing Tyrode’s solution. The tissues were maintained with a constant supply of carbogen gas (95% O_2_ + 5% CO_2_) [22]. Pieces of about 1.5 cm were cut from the jejunum and mounted in organ bath containing Tyrode’s solution already maintained with carbogen gas (95% O_2_, 5% CO_2_) at 37 ± 1 °C [23]. Test concentrations of Schiff base (SB1 and SB2) were prepared as mentioned. Schiff bases (SB1 and SB2) were added in a cumulative manner to the organ bath in test concentrations (10^−8^–10^−2^ M). Tissues were stabilized for a period of 30 min before assessing the Schiff base (SB1 and SB2). The test samples were applied periodically, at 2 min gaps. According to our documented protocols, the effects on spontaneous jejunal preparations were recorded in another series of experiments. We also tested the test samples on 80 mM KCL-induced contractions for possible relaxant effects that follow inhibition of voltage-gated calcium-channel-blocking activity [24,25].

### 2.7. Effects of Schiff Bases (SB1 and SB2) on KCl (80 mM)-Induced Contractions in Aortae

The experimental method employed was a modified version of the technique described by Furchgott and Bhadrakom (1953) [26,27]. We chose rabbits weighing 1.5–2 kg for this study, regardless of their sex. After euthanizing the rabbits, their thoracic cavities were exposed, and the surrounding blood vessels were carefully moved aside to isolate the thoracic aorta. The extracted aorta was then immersed in Krebs’ solution, which was continuously oxygenated using carbogen gas. Extra tissues and fats were removed from aortae under a dissecting microscope. Strips of aortae were soaked in Krebs’ solution. A sharp razor blade was used to cut the aortae into 2–3 mm strips. Aortic strips with intact endothelium were separated. To denude the aortae, a slightly moist cotton swab was gently rubbed over the luminal surface into the aortic strips. Denudation of aortae was confirmed by loss of response to acetylcholine (ACh) relaxation [28,29]. The aortae were secured between two triangular stainless-steel supports in a bath containing 15 mL of Krebs’ solution. The bath environment was regulated at 37 ± 1 °C, with a gas mixture of 95% oxygen and 5% carbon dioxide. An initial tension of 2 g was applied as a preload to the tissues.

For 40–60 min, the tissues were stabilized. KCl (80 mM) was applied directly to the aortae to induce sustained contractions [30]. After obtaining sustained aortic contractions, different molar concentrations (1 × 10^−8^–1 × 10^−2^ M) of SB1 and SB2 were applied in a cumulative manner with a 10 min gap. According to our documented methods, the effects of Schiff bases (SB1 and SB2) on KCl-induced contractions were noted [31,32,33]. Mean effective concentration (EC_50_) for the Schiff bases (SB1 and SB2) were calculated [34].

### 2.8. Effects of Schiff Bases (SB1 and SB2) on NE (1 µM)-Induced Contractions in Aortae

Both intact and endothelium-removed aortae were suspended in an organ bath continuously supplied with carbogen gas. The aortic tissues were allowed to stabilize for 40–60 min. To induce sustained aortic contractions, 1 µM of norepinephrine solution was added to the organ baths. Following peak contractions, Schiff bases (SB1 and SB2) were administered to the organ bath in test concentrations (1 × 10^−8^ to 1 × 10^−2^ M) to examine their potential effects. A 10 min interval was maintained between the applications of different test concentrations. The effects on NE-induced contractions were documented according to previously established protocols [31,32,35]. Mean effective concentrations (EC_50_) of Schiff bases SB1 and SB2 were calculated from the dose response curves. Each experiment was conducted at least three times.

### 2.9. Effects of Schiff Base SB1 and SB2 on NE-Induced Contractions of Denuded Aortae in the Presence of Verapamil

In a subsequent set of experiments, NE (1 µM) was used to induce contractions in endothelium-removed aortae. Verapamil (1 µM) was administered before NE application, with a 40 min incubation period. Schiff base compounds SB1 and SB2 were then applied cumulatively in concentrations ranging from 10^−8^ to 10^−2^ M. The responses were documented. The EC_50_ values were determined for the combined impact of Schiff bases SB1 and SB2 with verapamil on denuded aortae. Each experiment was conducted at least three times [27]. Any potential shifts in EC_50_ values were noted to identify drug interactions.

### 2.10. Effects of Schiff Bases (SB1 and SB2) on Norepinephrine Concentration Response Curves (N. ECRCs) in Aortae

SB1 and SB2 produced their effects mostly through the α1 adrenergic receptor blocking phenomenon. Therefore, to verify the involvement of alpha 1 adrenergic receptors in the aortae, we generated NE curves across a range of NE concentrations (1 × 10^−4^–256 × 10^−4^ M) with and without various concentrations of Schiff bases (SB1 and SB2). In brief, denuded aortic tissues were kept in Krebs’ solution at 37 ± 1 °C and stabilized at baseline. Control NE curves were first constructed without Schiff bases. After washing the tissues with Krebs’ solution, NE curves were then produced in the presence of different Schiff bases (SB1 and SB2) and concentrations after a 1 h incubation period. For comparison, we also constructed NE curves with and without terazosin, a known alpha one receptor blocker. The NE curves of Schiff bases were examined for potential right shifts and compared with the effects of terazosin [31,32].

### 2.11. Molecular Docking Analysis of Schiff Bases (SB1 and SB2)

In molecular docking analysis, Pyrex 0.8 software was used. It generated the top nine docked complexes in the cases of SB1 and SB2 with target receptor Gq 1D (P25100), based on the lowest binding energy score and root mean square deviation. In the case of SB1 and SB2 target receptors, −8.5 kcal/mole binding energy and score were predicted. Using BIOVIA Discovery Studio Visualizer bioinformatics software version 4, the interactive amino acid residues and binding clefts were analyzed [36].

### 2.12. Statistical Analysis

We calculated means and standard deviations using Microsoft Excel. Concentration response curves were drawn for both Schiff bases, with test concentrations on the *X*-axis and percent responses of control max (KCl and NE) on the *Y*-axis. Graph Pad Prism version 8 was employed to determine EC_50_ values. For N. ECRCs, we plotted responses against log NE molar (M) concentrations. One-way ANOVA with a 95% CI and *p* < 0.05 was used to compare EC_50_ values to their corresponding control EC_50_.

## 3. Results

SB1 and SB2 produced no effects on isolated jejunal preparations. However, the Schiff bases SB1 and SB2 produced significant effects on the isolated aortic strips’ preparations, respectively presented in Figure 2 and Figure 3.

Figure 2 illustrates the effects of Schiff base SB1 in isolated aortic strips (both denuded and intact) on KCL (80 mM) and norepinephrine (1 µM)-induced contractions.

Figure 3 displays how Schiff base SB2 works in isolated aortic strips (both denuded and intact) on contractions elicited by 80 mM KCl and 1 µM norepinephrine.

Respective EC_50_ values for Schiff base SB1 and Schiff base SB2, in endothelium-removed and intact aortae, are shown in Table 1.

As mentioned in Table 1, the Schiff base SB2 causes relaxation of NE-induced contractions for endothelium-removed and intact aortae, but it does not cross the EC_50_ line. However, in the case of Schiff base SB1, it does cross the EC_50_ line, so we cannot compare its potency to bases of EC_50_. We can say that Schiff base SB1 is more efficacious than Schiff base SB2 as it causes more relaxation expressed as a % of the NE control maximum in both endothelium-removed and intact aortae.

The relaxing effects of Schiff bases (SB1 and SB2) on norepinephrine-induced contractions pretreated with verapamil (1 µM) in endothelium-removed aortae are shown in Figure 4.

The additive or synergistic effects of verapamil and SB1 and SB2 are shown in Table 2.

Both Schiff bases (SB1 and SB2) have an additive or synergistic effect with verapamil on NE-induced contractions of denuded aortae because they caused more relaxation with verapamil.

N. ECRCs drawn for the confirmation of α1 receptors’ inhibition by Schiff bases (SB1 and SB2) are shown in Figure 5.

EC_50_ values for Schiff bases SB1 and SB2, derived from N. ECRCs (Figure 5), are shown in Table 3.

Schiff base SB1 in test concentration (9.1 × 10^−8^ M) caused a right shift in EC_50_ = −3.38 ± 0.03 (log NE) M vs. its respective control EC_50_ = −3.64 ± 0.00 (log NE). Similarly, Schiff base SB2 in test concentration (3 × 10^−8^ M) caused a right shift in EC_50_ = −2.58 ± 0.02 (log NE) M vs. its respective control EC_50_ = −3.51 ± 0.00 (log NE) M. Schiff bases SB1 and SB2’s right shift of EC_50_ values resembles terazosin’s right shift (a common alpha one receptor antagonist).

### 3.1. Molecular Docking and Docked Confirmation Visualization Analysis of Schiff Base SB1 and Schiff Base SB2

The top hit of the docking analysis revealed that the compounds can properly bind with the target receptor Gq 1D (P25100), and these compounds can be further stimulated to validate the binding stability under various pressures and dynamic environments. Over the docking pose, predicted binding energy and rmsd values are presented in Table 4.

### 3.2. Docked Complex Visualization of SB1

In the case of SB1 and receptor LEU187, ILE184, CYS180, THR181, PHE357, PHE263, PHE365, and TRP361 amino acids were reported to have been involved in the binding interaction through Van der Waals, unfavorable bump, pi-sigma, pi-pi shaped, alkyl, and pi-alkyl types of interaction have been observed, as mentioned in Figure 6.

### 3.3. Docked Complex Visualization of SB2

In the SB2 and receptor docked complex SER183, ASN398, VAL353, VAL354, VAL356, ASN394, PHE357, LEU187, ILE184, LEU138, TRP361, VAL358, and CYS360 amino acids residues were observed in interaction though Van der Waals, unfavorable bump, pi-lone pair, alkyl, and pi-alkyl interaction have been observed, as presented in Figure 7.

## 4. Discussion

It is pertinent to mention that the vasorelaxant phenomenon is multifaceted. Mostly, vasorelaxation follows the inhibition of voltage-gated calcium channels. Schiff base SB1 and Schiff base SB2 have not relaxed KCl (80 mM)-induced contractions of denuded and intact aortae, which implies that voltage-gated L-type calcium channels are not inhibited by these Schiff bases, as the relaxing effect on high molar (80 mM) KCL-induced contractions implies a vasorelaxing effect via L-type Ca^2+^ channels [27,37]. Thus, we adopted other protocols for the involvement of the inhibition of receptor-operated calcium channels. Consequently, Schiff bases SB1 and SB2 have relaxing effects on NE-induced contractions, which indicates an alpha-one-receptor-operated calcium-channel-blocking effect that contributes to the controlled release of calcium from internal reserves. Therefore, it may be stated that these Schiff bases (SB1 and SB2) are both strong alpha one receptor antagonists.

The Schiff base SB2 causes relaxation of 1 µM norepinephrine-induced contractions for endothelium-removed and intact aortae, but it does not cross the EC_50_ line. In case of the Schiff base SB1’s relaxing effect, it crosses the EC_50_ line and is relaxed up to 48% and 41% relative to the control in intact and denuded aortae, respectively. We can say that Schiff base SB1 is more potent than Schiff base SB2 as it causes more % relaxation of the NE control maximum, both in endothelium-removed and intact aortae. It is also clear from the results that Schiff base SB1 is more potent than Schiff base SB2 as it started relaxation at extremely low doses as compared to Schiff base SB2.

However, a further mechanism for the relaxing effect was confirmed through the concentration response curves with norepinephrine.

To look into the mechanisms of the relaxant effects of Schiff bases SB1 and SB2, their relaxing effects were studied on NE-induced contractions of denuded aortae pretreated with verapamil (a standard calcium channel blocker).

As shown in Table 2, the Schiff base SB1 demonstrates the highest efficacy, leading to 12.5% relaxation of norepinephrine-induced contractions pretreated with verapamil. Schiff base SB2 follows as the second most effective, causing 21.2% relaxation. It is clear that Schiff bases SB1 and SB2 both have additive or synergistic effects with verapamil on norepinephrine-induced contractions in denuded aortae because they cause more relaxation (SB1, 12.5% and SB2, 21.2%) with verapamil as compared to effect of the Schiff bases alone (SB1, 41% and SB2, 74%). These effects can be translated in many ways. For example, research has demonstrated that alpha-one-blockers have a minor but beneficial impact on the lipid profile of hypertensive patients [38]. This stands in contrast to the negative effects observed with high-dose diuretics and beta-blockers [39]. The growing recognition of hypertension as a cardiovascular risk factor, coupled with the understanding that combination therapy can enhance efficacy and tolerability in patients whose blood pressure is not adequately managed with single-drug treatment, has sparked interest in exploring new supplementary approaches for hypertension management. Studies clearly indicate that supplementing existing antihypertensive treatment with an alpha-one-blocker is an effective method to enhance blood pressure control in patients with poorly managed hypertension, regardless of their current medication regimen. Moreover, adding alpha-one-blockers to an existing treatment plan is well tolerated. Certain patient groups may find this approach particularly beneficial, including those with a metabolic profile associated with high cardiovascular risk and individuals with coexisting conditions such as type 2 diabetes mellitus, benign prostatic hyperplasia, chronic obstructive pulmonary disease, asthma, renal disease, and peripheral vascular disease [40].

The relaxing effects of Schiff base SB1 and Schiff base SB2 through inhibition of alpha-one-receptor-blocking activity on NE-induced contractions may play a crucial role in addressing vasoconstriction caused by stress or in circumstances where adrenaline levels spike. Thus, these molecules can be of importance in stress related to catecholamine surge.

The relaxing effects on contractions induced by norepinephrine required additional evidence confirming the inhibition of receptor-mediated calcium channels. For further confirmation of Schiff bases’ (SB1 and SB2) effects on alpha one receptors, N. ECRCs were constructed both in the absence (control curve) and presence of different concentrations of the tested Schiff bases (SB1 and SB2). A right shift in the EC_50_ (Log N. E) M for N. ECRCs implies that Schiff base SB1 and Schiff base SB2 have alpha one receptor blocking activity. Schiff base SB2 in test concentration 3 × 10^−8^ M produced a right shift in EC_50_ with a lower EC_50_ (−2.58 ± 0.02) log NE M, suggesting that Schiff base SB2 is more potent than Schiff base SB1. The EC_50_ right shift in N. ECRCs endorses the alpha one receptor inhibitory effects for the tested Schiff bases (SB1 and SB2). Alpha-1 blockers are multifunctional drugs employed in diverse medical contexts, such as preventing postoperative urinary retention (POUR) [41], addressing benign prostatic hyperplasia (BPH) [42] and lower urinary tract symptoms (LUTS) in females [43], aiding ureteric stone expulsion [44], and treating premature ejaculation [45]. While efficacious, these medications come with a spectrum of adverse effects, primarily linked to their influence on blood pressure and sympathetic nervous system activity. More recently developed, selective agents offer enhanced safety profiles, particularly for individuals with coexisting cardiovascular diseases [46].

When we correlated these effects with molecular docking investigations of the test substances, it was interesting to see that these molecules interacted with adrenergic receptors, which may virtually justify the results that were observed. In molecular docking analysis, Pyrex 0.8 software generated the top nine docked complexes in the cases of SB1 and SB2 with a target receptor based on the lowest binding energy score and root mean square deviation. In the case of SB1 and the target receptor, −8.5 kilocalorie per mole (kcal/mol) binding energy was predicted, while in the case of SB2 and the target receptor, the same binding score, −8.5 kcal/mol, was calculated. The top hit of the docking analysis revealed that the test molecules can properly bind with the target receptor. Hence, these compounds can be further studied to validate binding stability under various pressures and dynamic environments. Over the docking pose, predicted binding energy, and rmsd values, it is postulated that test molecules have selectivity to α1-adrenergic receptors. This predicted response was translated in isolated aortae where the test molecules showed alpha one receptor blocking activity, which was confirmed by constructing N. ECRCs in the presence and absence of SB1 and SB2 while using terazosin as a standard alpha-one-receptor-blocking agent.

## 5. Conclusions

The relaxing effects of Schiff base SB1 and Schiff base SB2 on NE-induced contractions are mediated via alpha one receptor blocking activity, which was confirmed by constructing N. ECRCs in the presence and absence of SB1 and SB2 while using terazosin as the standard alpha-one-receptor-blocking agent. Molecular docking also supports the affinity for alpha one receptor Gq 1D (P25100). This confirms that SB1 and SB2 lead molecules produce vasodilation via inhibition of alpha one adrenergic receptors. The Schiff bases SB1 and SB2 molecules interact additively with verapamil to produce more vasodilation response.

## 6. Recommendations

These Schiff bases, SB1 and SB2, can be considered as good options for further investigations exploring their detailed mechanisms.

## Figures and Tables

**Figure 1 biomolecules-15-00611-f001:**
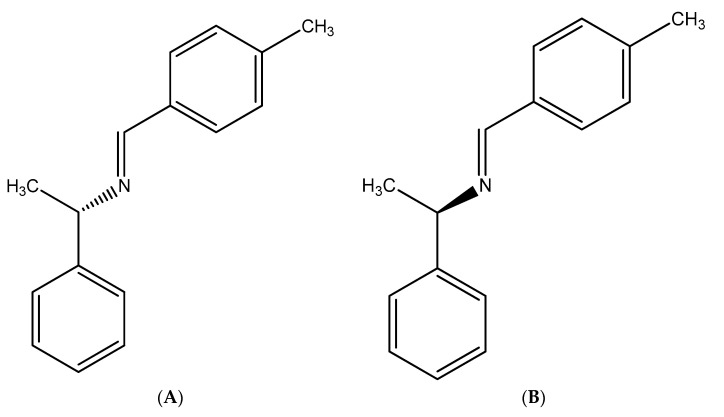
(**A**) Structural formula of SB1. (**B**) Structural formula of SB2.

**Figure 2 biomolecules-15-00611-f002:**
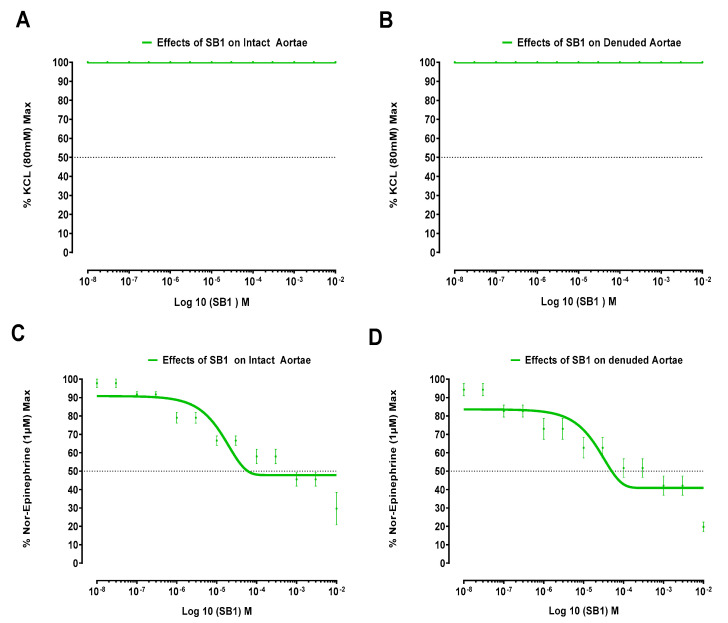
Impact of Schiff base SB1 on intact and endothelium-removed aortic tissue samples. For Schiff base SB1: (**A**) influence on potassium-chloride-induced contractions in intact aortic segments; (**B**) influence on potassium-chloride-induced contractions in endothelium-removed aortic segments; (**C**) influence on norepinephrine-induced contractions in endothelium-intact aortic segments; (**D**) impact on contractions induced by norepinephrine in endothelium-removed aortic segments. Data presented as mean ± SD (*n* = 3–4, *p* < 0.05).

**Figure 3 biomolecules-15-00611-f003:**
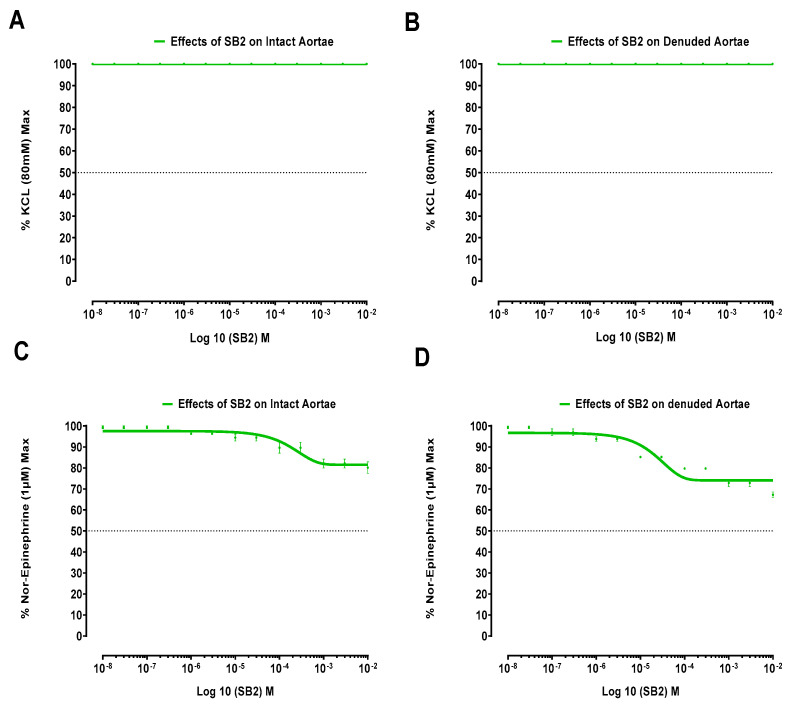
Impact of Schiff base SB2 on intact and endothelium-removed aortic tissue samples. For Schiff base SB2: (**A**) influence on potassium-chloride-induced contractions in intact aortic segments; (**B**) influence on potassium-chloride-induced contractions in endothelium-removed aortic segments; (**C**) influence on norepinephrine-induced contractions in intact aortic segments; (**D**) influence on norepinephrine-induced contractions in endothelium-removed aortic segments. Data presented as mean ± SD (*n* = 3–4, *p* < 0.05).

**Figure 4 biomolecules-15-00611-f004:**
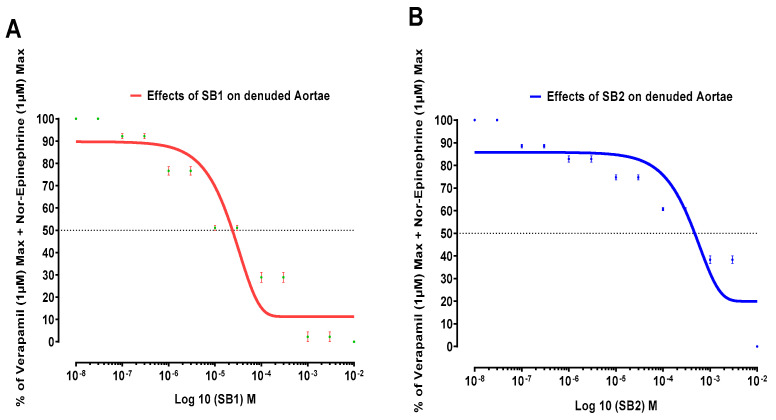
The impact of Schiff base SB1 (**A**) and SB2 (**B**) on norepinephrine-induced contractions in endothelium-removed aortic strips, with verapamil present, is examined. All data shown as mean ± SD (*n* = 3–4, *p* < 0.05).

**Figure 5 biomolecules-15-00611-f005:**
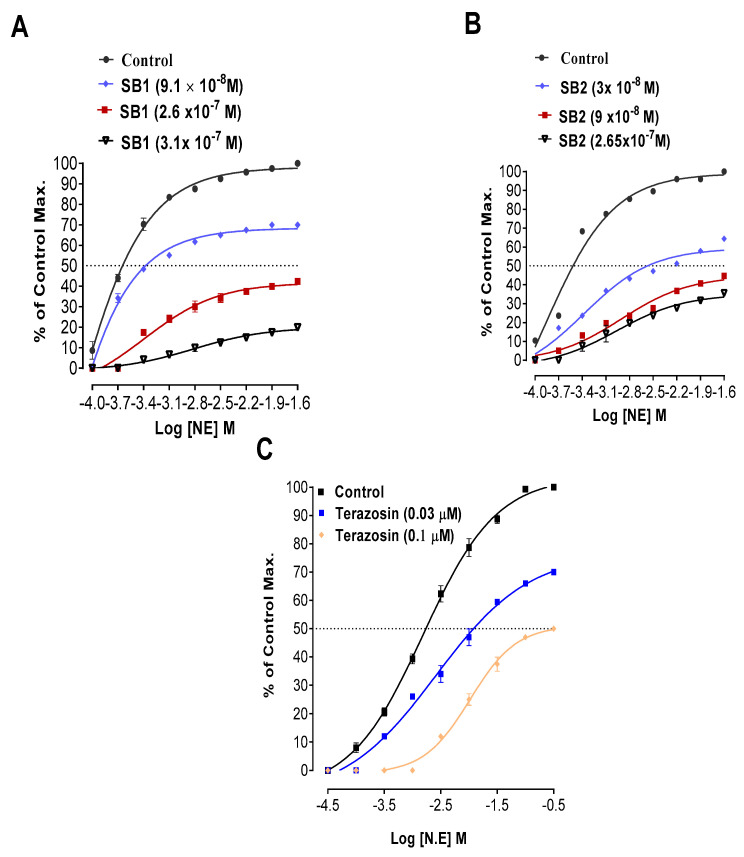
An analysis of how Schiff base SB1, Schiff base SB2, and Terazosin influence Norepinephrine Concentration Response Curves. (**A**) N. ECRCs with and without various concentrations of Schiff base SB1 in endothelium-removed aortae. (**B**) N. ECRCs with and without different concentrations of Schiff base SB2 in denuded aortae. (**C**) N. ECRCs with and without varying concentrations of Terazosin in endothelium-removed aortae. All data expressed as mean ± SD (*n* = 3–4).

**Figure 6 biomolecules-15-00611-f006:**
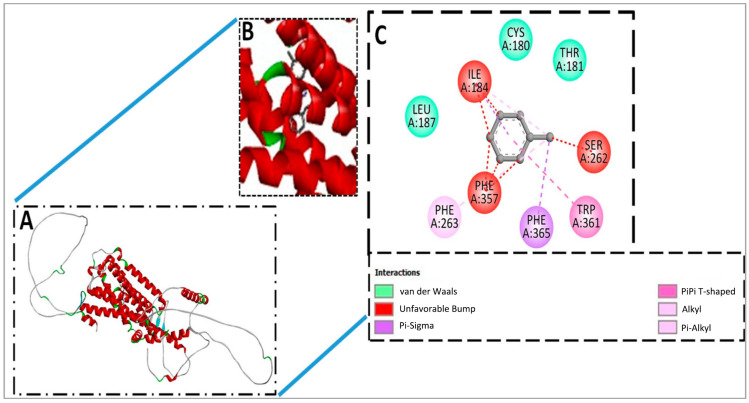
(**A**) Represented docked complex of SB1 and target receptor. (**B**) Unveiling the binding cleft of receptor Gq 1D (P25100) and attached compounds (SB1). (**C**) Interactive amino acid residues and several types of bonding.

**Figure 7 biomolecules-15-00611-f007:**
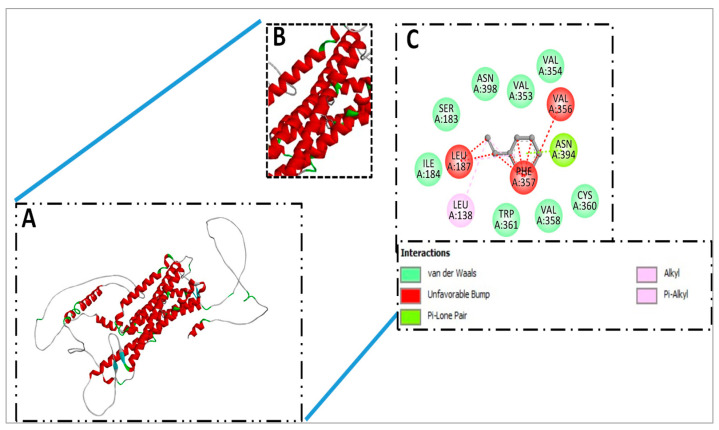
(**A**) Represented docked complex of SB2 and target receptor. (**B**) Unveil the binding cleft of receptor Gq 1D (P25100) and attached compounds (SB2). (**C**) Interactive amino acid residues and several types of bonding.

**Table 1 biomolecules-15-00611-t001:** To demonstrate Schiff base SB1 and Schiff base SB2‘s relaxing effects with their respective EC_50_ values (all values are mean ± SD, *n* = 3–4).

Test Compounds	Status of Aortae	% of Potassium Chloride (Control Max)		% of Norepinephrine (Control Max)	EC_50_ ± SD Potassium-Chloride-Induced (Molar)	EC_50_ ± SD Norepinephrine-Induced (Molar)
Schiff base (SB1)	Intact	No relaxation		48%	Nil *	5.50 × 10^−5^ ± 2.23
	Denuded	No relaxation		41%	Nil *	5.81× 10^−5^ ± 3.80
Schiff base (SB2)	Intact	No relaxation		82.5%	Nil *	Nil
	Denuded	No relaxation		74%	Nil *	Nil

* No EC_50_, as no relaxation occurred with KCL (80 mM).

**Table 2 biomolecules-15-00611-t002:** An investigation into the relaxation effects of Schiff bases (SB1 and SB2) on NE (1 µM)-triggered contractions in denuded aortae, both alone and with verapamil, including their respective EC_50_ values. All data presented as mean ± SD (*n* = 3–4).

Test Compounds	Status of Aortae	% of Norepinephrine (Control Max)	EC_50_ ± SD Norepinephrine-Induced (Molar)
Schiff base SB1	Denuded	41%	5.81 × 10^−5^ ± 3.80
Schiff base SB1 + verapamil	Denuded	12.5%	3.74 × 10^−3^ ± 0.00
Schiff base SB2	Denuded	74%	Nil **
Schiff base SB2 + verapamil	Denuded	21.2%	4.54 × 10^−4^ ± 0.00 M.

** Relaxation occurred with NE (1 µM), but it did not cross the EC_50_ line.

**Table 3 biomolecules-15-00611-t003:** To illustrate the right shift of EC_50_ of Schiff base SB1, Schiff base SB2, and terazosin on N. ECRCs. All values are mean ± SD (*n* = 3–4).

Schiff Bases	N. ECRCs Specifications	EC_50_ Log (NE) ± SD
Schiff base SB1	Control Test concentration (9.1 × 10^−8^ M) Test concentration (2.6 × 10^−7^ M) Test concentration (3.1 × 10^−7^ M)	−3.64 ± 0.00 −3.38 ± 0.03 **** Nil **** Nil ****
Schiff base SB2	Control Test concentration (3 × 10^−8^ M) Test concentration (9 × 10^−8^ M) Test concentration (2.65 × 10^−7^ M)	−3.51 ± 0.00 −2.58 ± 0.02 **** Nil **** Nil ****
Terazosin	Control Concentration of the test (0.03 µM) Concentration of the test (0.1 µM)	−2.77 ± 0.06 −1.94 ± 0.10 **** −0.61 ± 0.15 ****

(****) shows very significant values (*p* < 0.0001) in a one-way ANOVA compared to their respective controls.

**Table 4 biomolecules-15-00611-t004:** Docking score of Schiff bases (SB1 and SB2) with target receptor Gq 1D (P25100).

S. No	Ligand	Binding Affinity	Rmsd/Ub	Rmsd/Lb
1	P25100__ADA1D_HUMAN_SB1	−8.5	0	0
2	P25100__ADA1D_HUMAN_SB1	−7.8	2.387	1.336
3	P25100__ADA1D_HUMAN_SB1	−7.6	6.66	1.185
4	P25100__ADA1D_HUMAN_SB1	−7.3	3.792	2.531
5	P25100__ADA1D_HUMAN_SB1	−7.3	2.084	1.587
6	P25100__ADA1D_HUMAN_SB1	−7.2	47.244	45.882
7	P25100__ADA1D_HUMAN_SB1	−7	33.922	29.324
8	P25100__ADA1D_HUMAN_SB1	−7	5.81	3.658
9	P25100__ADA1D_HUMAN_SB1	−6.8	46.818	45.721
10	P25100_ADA1D_HUMAN_SB2	−8.5	0	0
11	P25100_ADA1D_HUMAN_SB2	−7.6	6.656	1.239
12	P25100_ADA1D_HUMAN_SB2	−7.2	47.126	45.577
13	P25100_ADA1D_HUMAN_SB2	−7.1	2.117	1.214
14	P25100_ADA1D_HUMAN_SB2	−6.8	46.86	45.735
15	P25100_ADA1D_HUMAN_SB2	−6.8	29.78	26.775
16	P25100_ADA1D_HUMAN_SB2	−6.7	15.689	13.49
17	P25100_ADA1D_HUMAN_SB2	−6.6	6.714	2.314
18	P25100_ADA1D_HUMAN_SB2	−6.5	27.734	24.762

## Data Availability

PowerLab data sheets analyzed during this study are available with the corresponding author on formal/reasonable request.

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
