# Peer review of "New Lead Schiff Bases Predominantly Mediate Vasorelaxant Activity Through α1 Receptor Blocking Activity"

_biomolecules, 2025, doi:10.3390/biom15050611_

Round 1

Reviewer 1 Report

Comments and Suggestions for Authors

This paper titled New Lead Schiff Bases Predominantly Mediate Vaso-relaxant 2
activity through α1 receptor blocking activity focuses on two Schiff bases. The goal od reasearch was this study was examination of  potential relaxant effects on blood vessels and GIT smooth muscles  of Schiff baes (SB1 and SB2) on isolated rabbit aortic strips.

The authors wrote that  they synthesised these compounds , but there is no information about spectral properties and purity of these compounds. So, we have no relevant information on the purity of these compounds, etc. They only cited ref 16, but it is not enough. This causes me to be unable to assess whether the results presented are entirely correct or not. It has to be completed.

In the Results chapter Authors say: It is stated that SB1 and SB2 produced no effects on isolated jejunal preparations (data ot shown), but at the end of Introduction there is a statement: Thus, the current work is aimed to discover the underlying mechanisms of Schiff base SB1 and SB2 on smooth muscles of aortae and jejunal.." Therefore, the jejunal data studies should be placed in the Supplementary Materials, or the last sentence in the Introduction should be revised.

In my opinion, conclusions can be improved. The authors mention there were only docking results, but they did more. it would be advisable to broaden the conclusions.

References are properly cited. English needs corrections. The methods, which were used by the Authors, are sufficient.

The obtained results are interesting but paper before publication needs revision.

Comments on the Quality of English Language

English requaires improvement. It is hard to read and understand what Authors mean. 

Author Response

Title: New Lead Schiff Bases predominantly mediate Vaso-relaxant activity through α1 receptor blocking activity 

This paper titled New Lead Schiff Bases Predominantly Mediate Vaso-relaxant
activity through α1 receptor blocking activity focuses on two Schiff bases. The goal of research was this study was examination of potential relaxant effects on blood vessels and GIT smooth muscles of Schiff baes (SB1 and SB2) on isolated rabbit aortic strips.

Comment 1: The authors wrote that they synthesised these compounds, but there is no information about spectral properties and purity of these compounds. So, we have no relevant information on the purity of these compounds, etc. They only cited ref 16, but it is not enough. This causes me to be unable to assess whether the results presented are entirely correct or not. It must be completed.

Corrections:

Many thanks for the concern of the reviewer. We are now providing the necessary synthesis steps and related spectra as supplementary file as our other group synthesized and our group is responsible for the pharmacological screenings. Thus, we uploaded that as supplementary file. It is now mentioned in the manuscript as well.

Comment 2:

In the Results chapter Authors say: It is stated that SB1 and SB2 produced no effects on isolated jejunal preparations (data not shown), but at the end of Introduction there is a statement: Thus, the current work is aimed to discover the underlying mechanisms of Schiff base SB1 and SB2 on smooth muscles of aortae and jejunal..." Therefore, the jejunal data studies should be placed in the Supplementary Materials, or the last sentence in the Introduction should be revised.

Corrections:

This concern of the reviewer seems to be valid. But if we go to the experimental section, where lead molecules have been assessed in the experimental protocols. However, we are bound to present the results of the experimental work done. Since there was no effect on the jejunal preparations, so we wrote in a single sentence, in results section, that we observed on effects on the jejunal preparations. If we carry this to supplementary files, which does not need to be there as we observed no effects. So, simple narrative in a single statement is sufficient. Therefore, we request the reviewer to retain the statement to maintain the quality of the work.

Comment 3:

In my opinion, conclusions can be improved. The authors mention there were only docking results, but they did more. it would be advisable to broaden the conclusions.

Corrections: Many thanks, we revised now the conclusions as asked to cover the whole work.

Comment 4: References are properly cited. English needs corrections. The methods, which were used by the Authors, are sufficient.

Corrections:

We revised the manuscript as advised. Grammatical errors are removed.

Comment 4:

English requires improvement. It is hard to read and understand what Authors mean. 

Corrections:

The English is revised, and the messages are very clear.

I hope now you will give a waiver for publication of the manuscript.

Thanks

Prof Niaz Ali

Reviewer 2 Report

Comments and Suggestions for Authors

     Authors have submitted the Research Article entitled "New Lead Schiff Bases predominantly mediate Vaso-relaxant activity through aι receptor blocking activity" in the biomolecules. In this study, two Schiff bases (SB1 and SB2) were used to investigate potential relaxant effects on blood vessels and GIT smooth muscles. This study was interesting and could attract readers. Some results were clearly presented and the conclusion was supported by the experimental results. I recommend the publication of the paper in this Journal after minor revision. Please carefully consider the comments below and make the necessary revisions before resubmitting the work.

  1. The synthetic procedure of two Schiff bases was lacking.
  2. Schiff bases were characterized by FT-IR, NMR and MS. At the same time, the corresponding spectra were provided as the supporting information.
  3. The purity of the synthesized Schiff bases needed to be determined, which was very important for bioactivity evaluation.
Comments on the Quality of English Language

No

Author Response

Pointwise answers to the reviewers’ comments:

Title:

New Lead Schiff Bases predominantly mediate Vaso-relaxant activity through α1 receptor blocking activity 

Authors have submitted the Research Article entitled "New Lead Schiff Bases predominantly mediate Vaso-relaxant activity through alpha 1 receptor blocking activity" in the biomolecules. In this study, two Schiff bases (SB1 and SB2) were used to investigate potential relaxant effects on blood vessels and GIT smooth muscles. This study was interesting and could attract readers. Some results were clearly presented, and the conclusion was supported by the experimental results. I recommend the publication of the paper in this Journal after minor revision. Please carefully consider the comments below and make the necessary revisions before resubmitting the work.

Comment 1:

  1. The synthetic procedure of two Schiff bases was lacking.

Corrections:

Our other research group has synthesized the lead molecules in a series of steps. For simplicity, we are adding those synthesis steps for the SB1 and SB2 as supplementary files for easy reference. Regarding pharmacological screenings we have presented the work here. So, the information is now provided as supplementary file.

Comment 2:

  1. Schiff bases were characterized by FT-IR, NMR and MS. At the same time, the corresponding spectra were provided as the supporting information.

Corrections:

Now we have provided the related spectra and information as supplementary file.

  1. Comment 3:
  1. The purity of the synthesized Schiff bases needed to be determined, which was very important for bioactivity evaluation.

Corrections/ actions taken: It is now mentioned in the respective file and appropriate position in the manuscript.

I hope now you will give a waiver for publication of the manuscript.

Thanks

Prof Niaz Ali

Round 2

Reviewer 1 Report

Comments and Suggestions for Authors

After revision, a paper can be published. The manuscript is now complete with references, highlights, and additional data in supplementary matareials.